# (Z)-Endoxifen and Early Recurrence of Breast Cancer: An Explorative Analysis in a Prospective Brazilian Study

**DOI:** 10.3390/jpm12040511

**Published:** 2022-03-22

**Authors:** Thais Almeida, Werner Schroth, Jeanine Nardin, Thomas E. Mürdter, Stefan Winter, Solane Picolotto, Reiner Hoppe, Jenifer Kogin, Elisa Gaio, Angela Dasenbrock, Raquel Cristina Skrsypcsak, Lucia de Noronha, Matthias Schwab, Hiltrud Brauch, José Claudio Casali-da-Rocha

**Affiliations:** 1Clinical Oncology Department, Erasto Gaertner Hospital, Curitiba 81520-060, Brazil; thaisaalmeida@gmail.com (T.A.); elisa_gaio@yahoo.com.br (E.G.); angeladbrock@yahoo.com.br (A.D.); raquecristina@gmail.com (R.C.S.); 2Post Graduation Department, Pontifical Catholic University of Parana, Curitiba 80215-901, Brazil; lnno.noronha@gmail.com; 3Dr. Margarete Fischer-Bosch Institute of Clinical Pharmacology, University of Tuebingen, 70376 Tuebingen, Germany; werner.schroth@ikp-stuttgart.de (W.S.); thomas.muerdter@ikp-stuttgart.de (T.E.M.); stefan.winter@ikp-stuttgart.de (S.W.); reiner.hoppe@ikp-stuttgart.de (R.H.); matthias.schwab@ikp-stuttgart.de (M.S.); hiltrud.brauch@ikp-stuttgart.de (H.B.); 4Clinical Research Department, Erasto Gaertner Hospital, Curitiba 81520-060, Brazil; jnardin@erastogaertner.com.br; 5School of Health Science, UniBrasil, Curitiba 82820-540, Brazil; jprimon@erastogaertner.com.br; 6Pharmacy Department, Erasto Gaertner Hospital, Curitiba 81520-060, Brazil; solane.picolotto@gmail.com; 7German Cancer Consortium (DKTK) and German Cancer Research Center (DKFZ), Partner Site Tübingen, 69120 Tuebingen, Germany; 8Cluster of Excellence iFIT (EXC2180) “Image-Guided and Functionally Instructed Tumor Therapies”, University of Tuebingen, 69120 Tuebingen, Germany; 9Department of Clinical Pharmacology, University of Tübingen, 69120 Tuebingen, Germany; 10Department of Biochemistry and Pharmacy, University of Tübingen, 69120 Tuebingen, Germany; 11Department of Oncogenetics, Erasto Gaertner Hospital, Curitiba 81520-060, Brazil; 12Department of Oncogenetics, A. C. Camargo Cancer Center, São Paulo 01525-001, Brazil

**Keywords:** breast cancer, CYP2D6, tamoxifen, (Z)-endoxifen, early recurrence, survival, Brazil

## Abstract

Adherence to treatment and use of co-medication, but also molecular factors such as CYP2D6 genotype, affect tamoxifen metabolism, with consequences for early breast cancer prognosis. In a prospective study of 149 tamoxifen-treated early-stage breast cancer patients from Brazil followed up for 5 years, we investigated the association between the active tamoxifen metabolite (Z)-endoxifen at 3 months and event-free survival (EFS) adjusted for clinico-pathological factors. Twenty-five patients (16.8%) had recurred or died at a median follow-up of 52.3 months. When we applied a putative 15 nM threshold used in previous independent studies, (Z)-endoxifen levels below the threshold showed an association with shorter EFS in univariate analysis (*p* = 0.045) and after adjustment for stage (HR 2.52; 95% CI 1.13–5.65; *p* = 0.024). However, modeling of plasma concentrations with splines instead of dichotomization did not verify a significant association with EFS (univariate analysis: *p* = 0.158; adjusted for stage: *p* = 0.117). Hence, in our small exploratory study, the link between impaired tamoxifen metabolism and early breast cancer recurrence could not be unanimously demonstrated. This inconsistency justifies larger modeling studies backed up by mechanistic pharmacodynamic analyses to shed new light on this suspected association and the stipulation of an appropriate predictive (Z)-endoxifen threshold.

## 1. Introduction

The selective estrogen receptor modulator tamoxifen has been the first targeted drug treatment for more than 50 years for the control of tumor growth via the competitive inhibition of the estrogen receptor (ER) expressed in nearly 80% of breast cancers. Studies over the past few decades confirmed that long-term adjuvant treatment reduces disease mortality by 31% and recurrence by 50% [1,2]. Following the introduction of aromatase inhibitors (AI) that block the conversion of androgens to estrogen [3], tamoxifen continues to be widely used in premenopausal women and postmenopausal women who experience AI adverse events. Although highly effective, recurrence may occur in up to one third of women within 10 years after a standard 5-year treatment scheme [1]. Inter-patient variability of drug response is considered a possible mechanism of tamoxifen failure that has been attributed to intrinsic resistance [4,5], poor adherence to treatment [6,7], as well as drug–drug interactions [8]. To minimize tamoxifen failure, suitable predictors of response are in demand. Given the known plasma variability of the main active tamoxifen metabolite, (Z)-endoxifen, in patients treated with the same dose of tamoxifen (20 mg) per day, (Z)-endoxifen has been suggested as a putative predictor [9,10,11]. Its formation depends on cytochrome P450 (CYP) enzymes, of which the CYP2D6 converts the major metabolite N-desmethyl-tamoxifen to (Z)-endoxifen [12]. CYP2D6 enzymatic activity is highly variable and largely attributed to CYP2D6 gene haplotypes that are responsible for variable plasma (Z)-endoxifen levels [13,14]. Accordingly, CYP2D6 polymorphisms have been promoted as potential biomarkers, yet they only partially explain the variability of plasma (Z)-endoxifen concentrations [15]. Therefore, the clinical use of CYP2D6 genotyping as a predictor of tamoxifen response has been cautioned, mainly due to the lack of proof-of-concept in studies, which failed to uniformly reproduce the CYP2D6 outcome relationship resulting from methodological inconsistencies [16,17,18,19]. With LC MS/MS being a reliable analytical tool for the measurement of active drug metabolite concentrations, an alternative approach is to correlate (Z)-endoxifen plasma levels with clinical endpoints. In the absence of a proven optimal (Z)-endoxifen plasma level during adjuvant treatment, putative thresholds between 9 nM and 16 nM have been suggested by independent studies, above which higher benefit may be expected [9,10,11,15], and which could ad interim serve as a benchmark until prospective trial confirmation becomes available. Here, we apply this approach to patients from a Brazilian, prospective cohort (Tamoxifen Adjuvant Interferers Study; TAIS) [20], which we consider an explorative study for the investigation of a correlation between plasma (Z)-endoxifen levels and the risk of recurrence in early/luminal breast cancer. Despite the small size of the study, we engaged in this modeling exercise as there is an urgent need for prospective data that can provide guidance for the design of large clinical studies in the future.

## 2. Materials and Methods

### 2.1. Study Design, Patients, and Tamoxifen Treatment

The objective of this study was to evaluate whether plasma (Z)-endoxifen levels predicted early breast cancer (BC) events (recurrence or death) within 5 years, in patients receiving adjuvant tamoxifen treatment. The secondary aim was to evaluate whether (Z)-endoxifen levels were associated with clinical, pathological, and phenotypic CYP2D6 metabolism variables.

From a previously defined prospective cohort of 225 HR-positive, tamoxifen-treated consecutive breast cancer patients enrolled based on pathologically confirmed diagnosis of BC (adenocarcinoma) between April 2014 and June 2015 at the Erasto Gaertner Hospital, a national referral center for the treatment of cancer in Curitiba, Southern Brazil, we included those 149 patients with early BC and available plasma metabolite data in this current study. Inclusion criteria for the initial cohort were age ≥ 18 years, stages I–III BC according to the TNM classification [21], epithelial histology according to the World Health Organization classification [22], luminal molecular subtype, and assignment to adjuvant treatment with tamoxifen (20 mg/day) for ≥5 years. Tumors with estrogen receptor (ER) and/or progesterone receptor (PR) positivity and HER2 negativity with Ki67 expression ≤ 14% were classified as luminal A; those with ER and/or PR positivity, HER2 negativity, and Ki67 expression > 14% were classified as luminal B; luminal cases with HER2 overexpression (+3/+3 or fluorescence in situ hybridization positivity), regardless of Ki67 expression, were classified as the luminal HER molecular subtype. Exclusion criteria were age > 90 years, receipt of chemotherapy or other hormone therapy, previous malignancy, and inability to fill out the study questionnaires or follow the project schedules. Ethical approval was obtained from the Brazilian National Commission of Ethical Research (protocol No. 894.864). All patients provided written informed consent [20].

Patients were followed at 21-day intervals during the chemotherapy period and periodically underwent clinical examination to determine the treatment response. After the initiation of tamoxifen treatment, they were interviewed at 3, 6, and 12 months by a clinical pharmacist with a standardized questionnaire, and peripheral blood samples were collected for (Z)-endoxifen quantification. Subsequently, they were followed through medical consultation and routine physical examination every 4 or 6 months, according to the institution’s protocol. Mammography was performed annually, and complementary imaging examinations were performed upon the detection of suspected signs or symptoms of recurrence. The last follow-up date was April 2019. pCR was defined as the absence of residual invasive and in situ cancer upon hematoxylin and eosin evaluation of resected breast specimens and all sampled regional lymph nodes following completion of neoadjuvant therapy.

The daily dose of 20 mg tamoxifen was provided at no cost through the Brazilian Unified National Health System for the entire period of this study. At each 3-monthly hospital pharmacy visit for tamoxifen dispensing, the women’s symptoms, adherence, and possibility of drug interaction were assessed and the pharmacist provided orientation.

### 2.2. Assessment of Treatment Adherence and Drug Interactions

We previously described the influence of tamoxifen adherence and CYP2D6 pharmacogenetics on plasma (Z)-endoxifen concentrations in the full TAIS cohort [20]. In short, tamoxifen adherence was assessed using the Morisky Green Levine questionnaire [23], validated internally at our institution [24]. Scores on this questionnaire, administered by a clinical pharmacist during medication dispensation, reflect high, medium, and low degrees of adherence. For the current analysis, which is based on our previous results for adherence and (Z)-endoxifen levels, we allocated women with high and medium scores to a high adherence group and those with low scores to a low adherence group, as reported in Nardin et al. [20]. At each visit, all patients received guidance from the clinical pharmacist for the regular use of medications and to survey possible drug interactions.

### 2.3. CYP2D6 Polymorphism Genotyping and Quantification of Plasma (Z)-Endoxifen Levels

At the time of diagnosis, genomic DNA was obtained from peripheral blood mononuclear cells and genotyped for the detection of CYP2D6 variants (QIAamp DNA Mini Kit; Qiagen), as described previously [20]. From the genotypes (diplotypes) and a score assigned to each allele, CYP2D6 enzyme activity scores were determined. Metabolism phenotypes were determined according to these scores (0, poor metabolizer, PM; 0.5–1, intermediate metabolizer, IM; 1.5–2, normal metabolizer, NM; 3, ultra-rapid metabolizer, UM; Appendix A) [25]. Heparinized plasma samples were obtained at 3, 6, and 12 months after tamoxifen treatment initiation. Plasma (Z)-endoxifen levels were measured by liquid chromatography–tandem mass spectrometry, as described previously [14].

### 2.4. Survival Analysis

Event-free survival (EFS) was determined from the date of tamoxifen treatment initiation to the first documented recurrence (local or distant) or death due to the disease. Survival analyses were performed using Kaplan–Meier estimators and Cox proportional hazard (PH) regression with modeling of continuous predictors by natural cubic splines (smooth functions consisting of piecewise third-order polynomials). Here, three, four, and five knots were pre-tested for each continuous predictor and the model with the lowest Akaike information criterion (AIC) was chosen [26]. Linear spline modeling and conditional inference trees were applied for sensitivity analyses. For dichotomized analyses, three previously proposed thresholds of 9 nM [10], 14.15 nM [9], 15.8 nM [11], and an intermediate cutoff between the latter of approximately 15 nM (Z)-endoxifen were tested. All statistical tests were two-sided and the significance level was set to 5%. All analyses were performed using the IBM SPSS Statistics software (version 24.0. IBM Corporation, Armonk, NY, USA) as well as packages party_1.3-9 [26] and rms_6.1-1 of statistical software R-4.0.0 (www.r-project.org; accessed on 23 February 2021) [27,28].

## 3. Results

### 3.1. Patient Characteristics

Patient characteristics are given in Table 1. The median age was 51.5 (range, 28–82) years, and 74.5% of women were of self-declared white ethnicity. Based on age, 41.2% of the women were considered pre-menopausal (≤49 years), with the remainder being considered postmenopausal (>49 years). The most prevalent histological subtype was ductal carcinoma (*n* = 122 (81.9%)), and the most prevalent subtypes were luminal A (32%) and B (55%) (Table 1). CYP2D6 metabolizer status was NM in 61%, UM in 4.1%, IM in 32.2%, and PM in 2.7%. In summary, approximately 65% had a normal or excessive and 35% had an impaired CYP2D6 metabolizer phenotype. Based on pharmaceutical monitoring, only one participant was found to be using a strong CYP2D6 inhibitor concomitantly with tamoxifen. No case required a switch to an aromatase inhibitor. At 3 months, no case of low adherence was detected; at 12 months, the frequency of low adherence was 10.6% (*n* = 16), as reported previously [20].

### 3.2. Associations of (Z)-Endoxifen Levels with Clinical, Pathological, and CYP2D6 Phenotype Characteristics

To avoid adherence bias, and because of tamoxifen’s long (7-day) half-life in plasma, we focused the analysis on the completion of the third month of tamoxifen treatment, when plasma levels of (Z)-endoxifen reached a steady state. At 3 months, (Z)-endoxifen levels above a putative threshold of 15 nM, motivated by the literature [9,11] were detected in 112 (75.2%) patients.

(Z)-endoxifen levels did not differ significantly by clinical and pathological variables (Table 1). As expected, significant associations with CYP2D6 metabolism phenotypes were detected. In individual and grouped (PM + IM vs. NM + UM) comparisons, PM and IM phenotypes had lower median (Z)-endoxifen levels (7.7 nM and 16.3 nM, respectively) than patients with NM or UM phenotypes (27.6 nM, and 38.0 nM, respectively; *p* < 0.001).

### 3.3. Associations of Clinical and Pathological Variables and CYP2D6 Metabolism Phenotypes with Clinical Outcomes

In univariate Cox PH regression analysis, stage was strongly associated with EFS (Stage III vs. Stage I: HR = 10.38; 95% confidence interval (CI), 2.82–38.14; *p* < 0.001). EFS did not differ by age, molecular subtype, Ki67, CYP2D6 phenotype, previous chemotherapy, BMI, or tamoxifen adherence (Appendix A).

### 3.4. Associations of (Z)-Endoxifen Levels with Clinical Outcomes

Three previous publications proposed (Z)-endoxifen thresholds of 9 nM, 14.15 nM, and 15.8 nM to distinguish patients at higher and lower risk for breast cancer recurrence or death [9,10,11]. Motivated by this, we observed significantly shorter EFS with low (Z)-endoxifen levels when using the two thresholds at 14.15 and 15.8 nM, whereas the association was not significant using the 9 nM cutoff (Appendix A). Thus, we applied an intermediate cutoff between the former of 15 nM in our cohort, revealing an association of (Z)-endoxifen levels below the threshold and shorter EFS rates in univariate Cox PH regression (HR = 2.27, 95% CI 1.02–5.06; *p* = 0.045; Figure 1), as well as after correction for stage (hazard HR = 2.52; 95% CI 1.13–5.65; *p* = 0.024; Table 2). However, since categorizing continuous variables has several disadvantages (e.g., loss of information), we also applied a modeling of (Z)-endoxifen levels with natural cubic splines. Here, we revealed no significant association between plasma concentrations and EFS in both univariate Cox PH regression (*p* = 0.158; Figure 2A) and after adjustment for stage (*p* = 0.117, Figure 2B). On average, spline modeling showed a more or less linear (but non-significant) decreasing risk for increasing (Z)-endoxifen levels up to a concentration range of 25–30 nM and an approximately constant risk for higher levels (Figure 2). However, as modeling with natural cubic splines implies smoothness of the estimated curve, jumps indicating intrinsic thresholds may have been discarded by this approach. Therefore, we performed sensitivity analyses using linear splines with knot positions chosen such that possible discontinuities within the range of 10–20 nM (Z)-endoxifen could be detected. However, here, and in conditional inference tree analysis, we did not find evidence for one or several cutoffs in our data set (data not shown).

## 4. Discussion

We investigated the association between steady-state active tamoxifen metabolite (Z)-endoxifen levels (at 3 months beyond tamoxifen initiation) and event-free survival (EFS) adjusted for stage in a small study from Brazil. Based on the prospective study design with the recruitment of patients with pathologically confirmed early breast cancer diagnosis, 5-year adjuvant tamoxifen treatment regimen (no switches to AI), and documentation of confounding factors such as treatment adherence and drug interactions, the study is predestined to address this relevant research question despite the currently enrolled and followed-up patients being limited. As this question is clinically highly relevant given the many patients embarking on long-term adjuvant tamoxifen treatment, we attempted an early statistical analysis and modeling approach to obtain first hints that may inform on the need for future study designs.

Using a putative clinical threshold concentration in the range between 14 and 16 nM, low plasma (Z)-endoxifen levels were associated with a higher rate of early recurrence or death events during follow-up. However, this finding was not supported when modeling (Z)-endoxifen on a continuous scale using regression splines, which has advantages as information loss due to categorization by thresholds is avoided [27]. On average, spline modeling showed a concentration/clinical effect dependency in the lower concentration range (<20–30 nM), whereas, in the higher concentration range (>25–30 nM), a (receptor) saturation plateau without any further clinical impact appears to have been reached. Although the trend of the curve was not significant and the patterns described above could have been observed simply by chance (Figure 1), the hypothesis of an optimal range for endoxifen is in line with previous findings suggesting that both low and very high endoxifen concentrations may promote recurrences [29]. Thus, larger modeling studies are required to shed new light on this suspected association and the stipulation of an appropriate predictive (Z)-endoxifen threshold.

In our previous investigation of this study cohort, we observed that, independently of CYP2D6 status, patient adherence to tamoxifen treatment significantly affected tamoxifen and (Z)-endoxifen levels, with regular intake leading to higher levels [20]. However, we could not observe an association with clinical outcome, likely due to the generally high compliance and low event rate as a result of the annual patient surveys. Likewise, despite the clear association of low active CYP2D6 metabolizer phenotypes (PM and IM) with low (Z)-endoxifen levels, CYP2D6 phenotype was not associated with clinical outcome. The unexpected finding of higher event rates in NM/UM compared to IM/PM (Appendix A) currently cannot be explained. As this discrepancy was mainly observed between NM and IM patients, it may be attributed to unknown CYP2D6 reduced-function alleles that escaped detection during standard CYP2D6 genotype-based phenotype assignment, thereby leading to a misclassification of NMs in this multi-ethnic Brazilian cohort.

A limitation of our study is the small study size with low power. According to statistical estimates, more than 3000 patients are required to detect a hazard ratio of 1.4 for patients with low (Z)-endoxifen [30]. However, its prospective design, unbiased selection, patient enrollment prior to the initiation of tamoxifen treatment, monitoring of adherence and drug interactions by clinical pharmacists, limited loss to follow-up (4.6% (*n* = 7)), and lack of switching to aromatase inhibitors provide us with solid grounds to use this cohort as a pilot for exploratory and hypothesis-generating research. Another limitation is the short follow-up time that is not fully informative on the number of events in luminal breast cancer, as the majority of recurrences will only occur beyond 10 years. Both the study size and duration of follow-up will gradually increase, as the study is ongoing and the cohort will be further followed up, with the option to re-analyze the data. Finally, we did not consider tumor-associated determinants of intrinsic endocrine resistance (e.g., variable effects of endoxifen on blocking transcription of ER-target genes involved in proliferation and migration) or acquired resistance, yet neoadjuvant studies with pre- and post-treatment biopsies, as well as longitudinal studies to survey the emergence of resistant ESR1 mutations, may offer future strategies.

## 5. Conclusions

Although we confirmed the possible use of previously suggested clinical thresholds, our study was not sufficiently large to either consistently prove or disprove an association between (Z)-endoxifen levels measured at the initial 3 months of treatment and EFS. Larger studies, including meta-analyses or utilizing neoadjuvant treatment windows, are required to shed further light on this important clinical research question. In addition, comprehensive modeling of an optimal therapeutic (Z)-endoxifen plasma concentration range together with mechanistic pharmacodynamic investigations may provide answers to whether and which (Z)-endoxifen plasma levels are of clinical value, and whether a risk associated with differences in plasma levels can be reduced by tamoxifen dose adaptations.

## Figures and Tables

**Figure 1 jpm-12-00511-f001:**
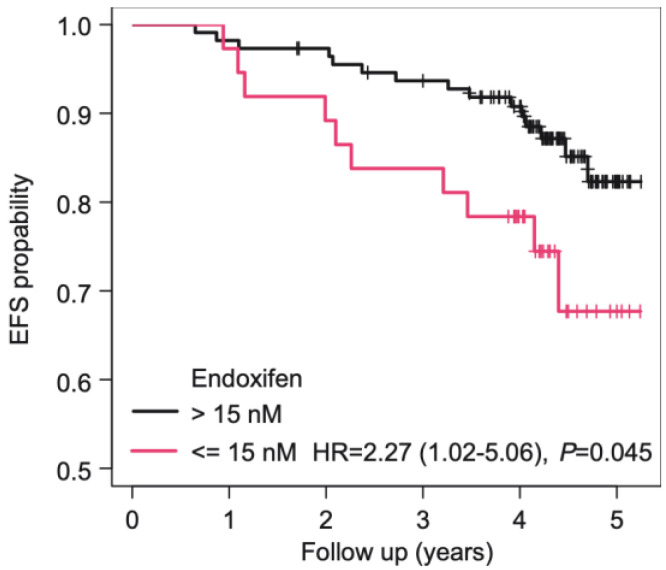
Kaplan–Meier curves of (Z)-endoxifen stratified by a putative threshold of 15 nM. The hazard ratio and 95% confidence intervals for below versus above threshold are given in brackets.

**Figure 2 jpm-12-00511-f002:**
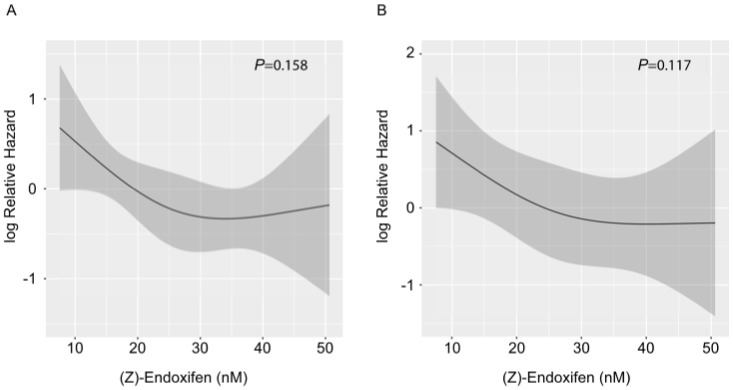
Cox proportional-hazards regression for EFS with modeling of (Z)-endoxifen levels using natural cubic splines (3 knots). (**A**) Univariate analysis, (**B**) multivariate analysis with adjustment for stage; 95% confidence levels are given as dark-grey shaded area.

**Table 1 jpm-12-00511-t001:** Patient, pathological, and CYP2D6 phenotype characteristics and their relations to (Z)-Endoxifen levels.

Characteristic	Category	Total (%)	(Z)-Endoxifen (nM)		*p* ^a^
			Median	Range	
Age (years)	>69	28 (18.9)	27.8	4.0–61.6	0.19
	>49–69	59 (39.9)	24.3	4.0–54.9	
	<49	61 (41.2)	24.4	4.4–48.8	
Ethnicity	White	111 (74.5)	24.9	4.0–67.0	0.08
	Black	9 (6.0)	18.8	10.1–40.8	
	Asian/Indian	4 (2.7)	38.7	41.5–61.6	
	Mixed-race	25 (16.8)	24.4	4.4–52.2	
BMI (kg/m^2^)	≤30	61 (70.1)	24.4	4.4–61.6	0.38
	>30	26 (29.9)	20.1	7.2–53.0	
Histology	Ductal carcinoma	122 (81.9)	24.7	5.7–54.9	0.98
	Lobular carcinoma	12 (8.1)	23.8	4.0–66.7	
	Others	15 (10.1)	24.9	4.4–61.6	
Staging	I	50 (33.6)	24.9	6.1–61.6	0.84
	II	74 (49.7)	24.4	4.0–52.6	
	III	24 (16.1)	28.6	5.7–67.0	
Molecular subtype	Luminal A	48 (32.4)	26.8	4.0–67.0	0.23
	Luminal B	82 (55.4)	24.3	6.3–61.6	
	HER2-positive	18 (12.2)	24.3	5.7–45.1	
Ki67 (%)	≤14	49 (33.1)	27.1	4.0–67.0	0.065
	>14	99 (66.9)	24.2	5.7–61.6	
Chemotherapy	Yes	94 (63.1)	24.7	4.4–67.0	0.75
	No	55 (36.9)	24.9	4.0–61.6	
Pathologic complete response	Yes	9 (16.4)	24.3	10.8–34.9	0.78
	No	46 (83.6)	25.1	4.4–67.0	
CYP2D6 phenotype class	PM	4 (2.7)	7.8	4.0–67.0	9.8 × 10^−6^
	IM	48 (32.2)	16.3	5.7–51.3	
	NM	91 (61.1)	27.6	4.4–54.9	
	UM	6 (4.0)	38.0	28.4–61.6	
CYP2D6 class combined	PM/IM	52 (34.9)	15.6	4.0–67.0	2.6 × 10^−6^
	NM/UM	97 (65.1)	28.2	4.4–61.6	

Data are presented as *n* (%). Abbreviations: SD, standard deviation; BMI, body mass index; HER2, human epidermal growth factor receptor 2; PM, poor metabolizer; IM, intermediate metabolizer; NM, normal metabolizer; UM, ultrarapid metabolizer. ^a^ Kruskal–Wallis test.

**Table 2 jpm-12-00511-t002:** Multivariate Cox proportional-hazard model of dichotomized (Z)-endoxifen and stage.

Factor	Level	Total (n)	Events [n (%)]	HR	95% CI	*p*
Tumor stage	I	50	3 (6)	Stage I vs. II: 0.33 Stage III vs. II: 3.71	0.09–1.17 1.59–8.72	0.0003
	II	74	12 (16.2)			
	III	23	10 (43.4)			
(Z)-endoxifen (nM) ^1^	≤15	36	10 (27.8)	≤15 vs. >15 2.52	1.13–5.65	0.024
	>15	111	15 (13.4)			

^1^ (Z)-endoxifen levels were dichotomized at >15nM based on literature cutoff. Abbreviations: HR, hazard ratio; CI, confidence interval.

## Data Availability

The data presented in this study are available on request from the corresponding author. The data are not publicly available due to ethical reasons.

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
