# Peer review of "(Z)-Endoxifen and Early Recurrence of Breast Cancer: An Explorative Analysis in a Prospective Brazilian Study"

_jpm, 2022, doi:10.3390/jpm12040511_

Round 1

Reviewer 1 Report

jpm-1632095

The reviewer’s report on

(Z)-Endoxifen and early recurrence of breast cancer: an explora-2 tive analysis in a prospective Brazilian study

Authors: Thais Almeida, Werner Schroth, Jeanine Nardin, Thomas Mürdter, Stefan Winter, Solane Picolotto, 4 Reiner Hoppe, Jenifer Kogin, Elisa Gaio, Angela Dasenbrock, Raquel Cristina Skrsypcsak, Lucia de Noro- nha, Matthias Schwab, Hiltrud Brauch , José Claudio Casali-da-Rocha*.

Comments to the authors

The authors evaluated firstly if plasma (Z)-endoxifen levels predict early breast cancer events (recurrence or death) within 5 years, in 149 patients receiving adjuvant tamoxifen treatment and secondly if plasma (Z)-endoxifen levels are associated with clinical, pathological, and phenotypic CYP2D6 metabolism variables. By using threshold of 15 nM (Z)-endoxifen, below which it showed an association with shorter event-free survival (EFS) in univariate analysis and after adjustment for stage. However, However, modeling of plasma concentrations with splines instead of dichotomization did not verify a significant association with EFS. The authors concluded that the link between impaired tamoxifen metabolism and early breast cancer recurrence could not be unanimously demonstrated in this small exploratory study.

Comments to the authors

In general, the manuscript is in good structured and logically presented. Whether the authors provide sufficient data to fully support the issues remains to be clarified

The critical issues are stated below.

  1. Association of (Z)-endoxifen levels with clinical outcomes have recently been studied broadly. The thresholds of endoxifen can vary from 16 nM (Madlensky et al., 2011 Pharmacol. Ther. 89, 718) to 3.26 nM (Helland et al., 2017 Breast Cancer Research 19,125). A small explorative study of 48 Asian women with breast cancer even demonstrated a J-shaped relationship in which patients with endoxifen levels <53.6 nM and >187.4 nM were at higher risk of recurrence (Love et al., 2013 SpringerPlus 2, 52). In the report, (Z)-endoxifen levels did not differ significantly by clinical and pathological variables (Table 1) but significant associations with CYP2D6 metabolism phenotypes were detected. It is known that the conversion of tamoxifen to the main active metabolite endoxifen is mediated by several cytochrome P450 (CYP) enzymes in which CYP2D6 plays a central role (Stearns et al., 2003 J. Natl. Cancer Inst. 95, 1758). However, recent review has shown that other cytochrome P450 (CYP) enzymes also have a role in the generation of the main active metabolite endoxifen (Helland et al., 2021 J. Pers. Med. 11, 201).

    Even though using 15 nM of (Z)-endoxifen as a threshold was based on previous study of the authors, the authors ought to perform the analysis setting with different (Z)-endoxifen thresholds, in which the association of (Z)-endoxifen levels with clinical outcomes may be revealed.

  1. It is known that the rationale for tamoxifen treatment of hormone receptor positive breast cancer is to block transcription of estrogen receptor (ER)-regulated genes involved in breast cancer differentiation, proliferation, and migration (Manavathi et al., 2013 Rev. 34, 1). Moreover, the mechanism of tamoxifen action entails joining the ligand binding pocket of the ER that promotes a conformational change preventing recruitment of coactivators and instead cause association with corepressors thus repressing transcriptional activity of the ER (Shiau, et al., 1998 Cell 95, 927; Brzozowski, et al., 1997 Nature 389, 753). Given to the above mentioned fact, the ER variants would be critical to the breast cancer differentiation, proliferation, and migration, hence the clinical outcomes.

Incorporating ER variants of patients into the analyses is able to clarify the associations of (Z)-endoxifen levels with clinical, pathological, and CYP2D6 phenotype characteristics. Additionally, ER-downstream targets’ variants of the patients apparently are equally critical in evaluation of the association of (Z)-endoxifen levels with clinical, pathological, and CYP2D6 phenotype characteristics.

  1. Minor: The abbreviated name “BC” in line 89 of the Materials and Methods section has not been shown the full name “breast cancer” when it first appears

Reviewer 2 Report

Introduction provides sufficient background and include all relevant references, but 7 of 30 of them are selfcitations.

Research design and methods are appropriate and are well-described.

Results are clearly presented and Discussion support them but there are not a Conclusion separated paragraph.

Perhaps it is also interesting make separated paragraphs (marked with bold font Introduction, Methods, Results...) in the Abstract, but not neccesary.
